# The Importance of Porins and β-Lactamase in Outer Membrane Vesicles on the Hydrolysis of β-Lactam Antibiotics

**DOI:** 10.3390/ijms21082822

**Published:** 2020-04-17

**Authors:** Si Won Kim, Jung Seok Lee, Seong Bin Park, Ae Rin Lee, Jae Wook Jung, Jin Hong Chun, Jassy Mary S. Lazarte, Jaesung Kim, Jong-Su Seo, Jong-Hwan Kim, Jong-Wook Song, Min Woo Ha, Kim D. Thompson, Chang-Ro Lee, Myunghwan Jung, Tae Sung Jung

**Affiliations:** 1Laboratory of Aquatic Animal Diseases, Institute of Animal Medicine, College of Veterinary Medicine, Gyeongsang National University, Jinju 52828, Korea; ksw0017@hanmail.net (S.W.K.); leejs058@gmail.com (J.S.L.); gladofls@naver.com (A.R.L.); wjdwodnr0605@gmail.com (J.W.J.); hilanamang@naver.com (J.H.C.); jassylazarte@yahoo.com (J.M.S.L.); afteru70@gmail.com (J.K.); 2Coastal Research & Extension Center, Mississippi State University, Starkville, MS 39567, USA; 9887035@hanmail.net; 3Environmental Chemistry Research Center, Korea Institute of Toxicology Gyeongnam Department of Environmental Toxicology and Chemistry, Jinju 52834, Korea; jsseo@kitox.re.kr (J.-S.S.); jjong@kitox.re.kr (J.-H.K.); songjk@kitox.re.kr (J.-W.S.); 4College of Pharmacy, Jeju National University, 102, Jejudaehak-ro, Jeju-si, Jeju-do 63243, Korea; beneminu_@naver.com; 5Moredun Research Institute, Pentlands Science Park, Penicuik, Midlothian EH26 0PZ, UK; kim.thompson@moredun.ac.uk; 6Department of Biological Sciences, Myongji University, Yongin, Gyeonggido 449-728, Korea; crlee@mju.ac.kr; 7Department of Microbiology, Research Institute of Life Sciences, College of Medicine, Gyeongsang National University, Jinju 52727, Korea; mjung@gnu.ac.kr; 8Centre for Marine Bioproducts Development, College of Medicine and Public Health, Flinders, University, Bedford Park, Adelaide, SA 5042, Australia

**Keywords:** outer membrane vesicles (OMVs), β-lactamase, porin, β-lactam antibiotic, *Escherichia coli*, hydrolysis

## Abstract

Gram-negative bacteria have an outer membrane inhibiting the entry of antibiotics. Porins, found within the outer membrane, are involved in regulating the permeability of β-lactam antibiotics. β-lactamases are enzymes that are able to inactivate the antibacterial properties of β-lactam antibiotics. Interestingly, porins and β-lactamase are found in outer membrane vesicles (OMVs) of β-lactam-resistant *Escherichia coli* and may be involved in the survival of susceptible strains of *E. coli* in the presence of antibiotics, through the hydrolysis of the β-lactam antibiotic. In this study, OMVs isolated from β-lactam-resistant *E. coli* and from mutants, lacking porin or β-lactamase, were evaluated to establish if the porins or β-lactamase in OMVs were involved in the degradation of β-lactam antibiotics. OMVs isolated from *E. coli* deficient in β-lactamase did not show any degradation ability against β-lactam antibiotics, while OMVs lacking OmpC or OmpF showed significantly lower levels of hydrolyzing activity than OMVs from parent *E. coli*. These data reveal an important role of OMVs in bacterial defense mechanisms demonstrating that the OmpC and OmpF proteins allow permeation of β-lactam antibiotics into the lumen of OMVs, and antibiotics that enter the OMVs can be degraded by β-lactamase.

## 1. Introduction

Since the discovery of penicillin, antibiotics have been responsible for prolonging human life and advancing human medicine. However, antibiotic-resistant bacteria, also known as superbugs or multi-drug resistant (MDR) bacteria, have emerged due to the indiscriminate misuse of antibiotics [1,2]. O’Neill (2014) has estimated that by 2050, 10 million deaths will have occurred each year as a result of antibiotic resistance and this is estimated to cost up to 100 trillion USD [3]. In 2013, the U.S. Centers for Disease Control and Prevention predicted that at least 2 million antibiotic-resistant bacteria infections, resulting in a predicted 23,000 deaths, would cost around 20 billion USD in extra healthcare, leading to an economic loss of at least 35 billion USD in the U.S. each year [4]. In 2016, the UK government reported that 700,000 deaths occur worldwide each year as a result of antibiotic-resistant bacteria [5]. This problem is not confined to humans, but spreads across species, affecting agriculture, livestock, fisheries, food and the environment [6]. Antibiotic-resistant bacteria are now regarded as the biggest challenge facing public health and efforts to reduce MDR bacteria globally have increased substantially.

All Gram-negative bacteria secrete spherical membrane bilayer structures (10 to 250 nm), referred to as outer membrane vesicles (OMVs), into the external environment during both in vitro growth and in vivo infection [7,8,9,10]. We now have a greater understanding of the composition, physicochemical properties and various roles of OMVs [8,10,11,12,13,14,15,16,17]. OMVs consist of outer membrane proteins, cytoplasmic proteins, periplasmic membrane proteins, phospholipids, lipopolysaccharides and genetic material [8,12,13]. More recent research has focused on the role of OMVs in protecting bacteria by directly participating in the bacteria’s development of antibiotic resistance [7,9,18,19,20,21]. However, there are few in-depth studies examining the mechanisms OMVs use to protect bacteria against antibiotics. Although many studies have investigated the effects of β-lactam antibiotics on bacteria, showing inactivation of β-lactamase and mutation of porin-encoding genes [22,23,24,25,26], the interaction between β-lactamases and porins in OMVs and β-lactam antibiotics remains to be clarified. Our previous work showed that OMVs from β-lactam-resistant *E. coli* can help β-lactam-susceptible *E. coli* avoid the effects of β-lactam antibiotics through hydrolysis. In addition, porins (OmpC and OmpF) and β-lactamase (Blc1) were seen to be upregulated in OMVs of β-lactam-resistant *E. coli* compared to OMVs of β-lactam-susceptible *E. coli* [7]. Therefore, we hypothesize that the increased number of porin proteins are able to efficiently direct the β-lactam antibiotics into the OMVs lumen, and the increase in β-lactamase actively drives the degradation of β-lactam antibiotics, suggesting that antibiotic hydrolysis is commonly observed in OMVs from β-lactam-resistant *E. coli* (RC85^+^) (Figure 1). In the present study, we attempt to demonstrate β-lactam antibiotic hydrolysis by OMVs by making mutants containing *ompC*, *ompF*, or *blc1* gene deletions and observing whether OMVs isolated from the mutants are able to consume β-lactam antibiotics within the bacterial environment and within a cell-free system.

## 2. Results

### 2.1. Characterization of Mutant Strains

To establish if Blc1, OmpC, or OmpF are involved in the OMVs’ ability to degrade β-lactam antibiotics, mutants were produced from RC85^+^ by knocking out each of these genes. The successful deletion of *blc1*, *ompC*, and *ompF* in mutant RC85^+^ strains was confirmed by PCR amplification shown in Appendix A. Mutant strains grew well in LB medium, having a logarithmic phase growth similar to RC85^+^ (Figure 2). The deletion of *blc1* and *ompF* had no distinguishable influence on growth rates, while the growth rate of ΔompC RC85^+^ was slightly slower than that of RC85^+^. When the growth on LB agar was observed, mutant strains formed smooth, slightly elevated, non-pigmented colonies, similar to those of RC85^+^ (data not shown). An antimicrobial sensitivity test was conducted with the mutant strains to determine whether changes in their antibiotic resistance occurred compared to RC85^+^ (Table 1). In the absence of the *blc1* gene, the minimum inhibitory concentration (MIC) of all β-lactam antibiotics was reduced. In the case of ΔompC, there was no difference in MIC levels relative to RC85^+^, apart from the MIC for cefazolin, which was enhanced, whereas inactivation of the *ompF* gene conferred more resistance to cefoperazone, cefazolin, and cefalexin in the mutant compared with RC85^+^.

### 2.2. Quantification of the Produced OMVs

The OMVs from the mutants and RC85^+^ were isolated after incubation under the same culture conditions. Electron micrograph analysis exhibited the similarity of OMVs isolated from the mutants and RC85^+^ in size with the spherical structure (Appendix A). The average diameter of the OMVs from the Δblc1, ΔompC, and ΔompF cells was nearly identical, while RC85^+^ OMVs were slightly larger than these (Appendix A). Production of OMVs was evaluated with a BCA protein assay, with the production of OMVs slightly decreased in Δblc1, but increased by 2.2- and 1.8-fold in ΔompC and ΔompF, respectively, relative to the level of OMVs produced by RC85^+^ (Figure 3).

### 2.3. Comparison of β-Lactamase Activity

Differences in β-lactamase activity between OMVs from RC85^+^ and the mutant strains were examined, based on a change in absorbance of OD_490_ over time (Figure 4a). Since nitrocefin can enter into bacteria through porins, the individual OMVs were destroyed by sonication to remove the variables for porins in OMVs. This liberates the β-lactamase present in the lumen of the OMVs. The absorbance obtained for mutant Δblc1 was similar to the negative control, while mutants ΔompC and ΔompF showed higher levels of absorbance than the positive control and they exhibited similar levels of β-lactamase activity to that of the RC85^+^ OMVs over the course of the experiment. The β-lactamase activity of the respective OMVs was expressed as milliunit per milligram (mU/mg) of OMV protein (Figure 4b). The β-lactamase activity of ΔompC and ΔompF OMVs was 72.4 mU/mg and 70.3 mU/mg respectively, nearly identical to those of RC85^+^ OMVs (64.4 mU/mg). OMVs from Δblc1 cells displayed the lowest β-lactamase activity of 2.7 mU/mg.

### 2.4. Evaluation of the Protective Role of OMVs against β-Lactam Antibiotics

To determine if the loss of porin or β-lactamase proteins from the OMVs influences the degradation of β-lactam antibiotics, we investigated the effect of OMVs from RC85^+^ and mutants (Δblc1, ΔompC, or ΔompF) on the growth of β-lactam susceptible *E. coli* (RC85) cells in the presence of a growth-inhibitory dose of six β-lactam antibiotics (Figure 5). When RC85^+^ OMVs were mixed with the antibiotics corresponding to a growth inhibitory concentration for RC85, the cells grew at the same or slower rate than the positive control (RC85 cells in LB medium without antibiotics). RC85 treated with OMVs from the ΔompC mutant grew in all antibiotics tested, but their growth was slower than the samples containing RC85^+^ OMVs. Furthermore, RC85 incubated with ΔompF OMVs grew after 24 h in cefoperazone and after 18 h in cefazolin, which was slower than that obtained with the ΔompC OMVs, while no growth was detected in the presence of the other four antibiotics (ampicillin, cefotaxime, amoxicillin, and cefalexin) over the 36 h culture period. On the other hand, RC85 incubated with Δblc1 OMVs did not show any growth when each of the six antibiotics was present. After the growth curve experiment (Figure 5), all samples were plated on nutrient agar with or without each of the antibiotics in the same concentration as was used in the growth curve experiment (data not shown). If the susceptible strains of *E. coli* (RC85) received antibiotic-resistant substances through OMVs during the experiment in Figure 5, it could grow on nutrient agar containing respective antibiotics. All samples that grew in the above experiment were grown in nutrient agar but not in nutrient agar with respective antibiotics. These results demonstrated that the survival rate of RC85 was not due to transfer of β-lactam resistant materials to RC85 by OMVs but was due to molecules owned by OMVs that protected the RC85 from the antibiotic environment. The colonies grown in nutrient agar were identified as *E. coli* at the species level using the MALDI-Biotyper (Bruker Daltonics, Bremen, Germany, data not shown).

### 2.5. Hydrolysis of β-Lactam Antibiotics by OMVs

Concentrations of β-lactam antibiotics were measured at specific time points in a cell-free system to determine whether β-lactam antibiotics could be hydrolyzed by the OMVs (Figure 6). Compared with the positive control, containing antibiotics without OMVs (0% hydrolysis), there were significant differences observed between OMVs from RC85^+^, Δblc1, ΔompC, and ΔompF in their ability to degrade the different β-lactam antibiotics tested. With all six β-lactam antibiotics examined, RC85^+^ OMVs showed the highest hydrolytic activity, followed by ΔompC then ΔompF OMVs, while no change in antibiotic concentration was noted with Δblc1 OMVs. These results imply that β-lactamase is the most important factor in the degradation of β-lactam antibiotics by RC85^+^ OMVs, alongside porin, specifically OmpF, which showed higher permeability to all six β-lactam antibiotics tested when compared to OmpC.

## 3. Discussion

We previously showed that porin proteins and β-lactamase enzyme are more abundant in OMVs isolated from β-lactam-resistant *E. coli* than from β-lactam-susceptible *E.coli*, and only OMVs from β-lactam-resistant *E. coli* were found to degrade β-lactam antibiotics [7]. Therefore, here we were interested in establishing whether the loss of porin or β-lactamase could directly influence the hydrolysis efficiency of OMVs, especially since the mechanism by which OMVs degrade β-lactam antibiotics is unknown. The aim of the present study was to establish what significance β-lactam antibiotic resistance-associated proteins, such as β-lactamase and porin, had on the production and activity of β-lactamase, and on the ability of OMVs from *E. coli* to degrade β-lactam antibiotics. Our results suggest that it is not the loss of β-lactamase but the loss of porin from the outer membrane of the OMV that influences the yield of OMVs obtained. The loss of porin does not affect the β-lactamase activity of OMVs but the loss of β-lactamase dramatically eliminated β-lactamase activity by the OMVs. Thus, the presence of β-lactamase and porin in the OMVs plays a significant role in the direct hydrolysis of β-lactam antibiotics.

Many studies have demonstrated that OMVs serve as a defense by the bacterium against antimicrobial peptides and antibiotics. For instance, OMVs from β-lactam-resistant *E. coli* plays an important role in the growth of susceptible bacteria by degrading β-lactam antibiotics before they can affect the bacteria [7]. OMVs containing β-lactamase enzymes inactivate some β-lactam antibiotics [7,19,20,27] or sequester some antibiotics [9], both leading to the protection of bacteria against corresponding antibiotics. OMVs can act as a vehicle for disseminating genetic material, including antibiotic resistance genes to susceptible bacteria, thereby contributing to the production of antibiotic-resistant bacteria [18,28,29]. These bacteria can protect susceptible bacteria by serving as decoys or acting as a physical shield, which helps them to evade the influence of some antibiotics [7,30,31,32]. Substances involved in antibiotic resistance are relatively safe from dilution and degradation because they are packed safely inside the OMVs [33]. Our results show that OMVs from RC85^+^ directly degrade β-lactam antibiotics to protect sensitive strains from antibiotic environments (Figure 5 and Figure 6).

Several studies have demonstrated that the loss of porins from the bacterial outer membrane can impact the production of OMVs. For example, Mcbroom et al. (2006) indicated that relative OMV production from an *E.coli ompC* mutant was significantly enhanced by almost 10-fold compared to the wild-type *E.coli* [34]. A deletion in *ompA*, encoding an outer membrane β-barrel protein with a periplasmic peptidoglycan-interaction domain resulted in a 26-fold hypervesiculation in *E. coli* mutant [35]. Valeru et al. (2014) showed a 3-fold increase in the level of production of OMVs by an OmpA mutant of *Vibrio cholerae* compared to the wild-type [36]. In line with these findings, our results showed that a lack of porins enhances the release of OMVs (Figure 3). The *E. coli* cells lacking porin proteins in their outer membrane (OmpC and OmpF) showed instability, with increased OMVs production due to a structural deficiency of the outer membrane [37]. Therefore, we speculate that a loss of porins alters the composition of the envelope membrane, which in turn affects membrane integrity, leading to enhanced secretion of OMVs.

β-lactam antibiotics are widely used antibiotics that are highly effective in combating bacterial infections [38]. These include penicillin derivatives, cephalosporins, monobactams, and carbapenems, and work by inhibiting cell wall biosynthesis, causing bactericidal effects for the bacteria. *E. coli* has developed four major mechanisms to resist the inhibitory effect of β-lactam antibiotics: inactivation of the antibiotics by enzymes, alteration of the active site of PBPs (penicillin-binding proteins), decreased permeation of the antibiotics and increased efflux of the antibiotics [39,40]. β-lactamases in the periplasmic space break the structure of the β-lactam ring, making the molecule’s antibacterial properties inactive so that antibiotics are unable to bind to PBPs [41]. Porin proteins produce transmembrane diffusion channels in the outer membrane that enable the diffusion of small hydrophilic molecules (e.g., sugars, amino acids, and vitamins) and β-lactam antibiotics to penetrate into the periplasmic space [42,43,44].

An observed decreased in the resistance of the β-lactamase mutants to β-lactam antibiotics compared with the wild-type was due to reduced β-lactamase activity [45,46]. Previous studies revealed that OmpC seems to be related to the transport of some β-lactam antibiotics [47,48,49,50]. Choi and Lee (2019) demonstrated that the OmpF-defective *E. coli* mutants showed increased resistance to several β-lactam antibiotics, such as ampicillin, cefalotin, cefoxitin, ceftazidime, aztreonam, and imipenem [47]. The absence of OmpF classical porin resulted in a significant increase in β-lactam resistance, including ampicillin and cefoxitin [51]. Our findings corroborate previous reports that the MIC against several β-lactam antibiotics was decreased or increased in single isogenic β-lactamase or porin mutants, respectively (Table 1). Based on the available data, we speculate that the change seen in MIC can be attributed to the reduced degradation of β-lactam antibiotics because of a lack of β-lactamase activity or decreased permeability of β-lactam antibiotics due to the absence of porin.

OmpC and OmpF are considered as the leading transport porins that assist penetration of most β-lactam antibiotics [47,51,52,53], and both porins are known to be major protein components of *E. coli* OMVs [54]. Diffusion rates through these channels differ according to a substance’s molecular weight and electrical charge [48,55]. Chemicals with hydrophilic molecules up to 600–700 Da in size can generally pass through the porin pores [56]. Among the six β-lactam antibiotics tested here (Figure 5 and Figure 6), ampicillin with the lowest molecular weight (349 Da) and cefoperazone with the highest (645 Da) were able to penetrate the pores of the OMVs. Compounds with one negative charged group (monoanionic compounds) penetrate porin channels faster than zwitterionic compounds [55]. Of the antibiotics tested, cefotaxime, cefoperazone, and cefazolin are monoanionic compounds and ampicillin, amoxicillin, and cefalexin are zwitterionic compounds.

The OmpF porin allows more efficient permeation of solute molecules than the OmpC porin channel in terms of the size of the channel, in particular, OmpF channel is 7% to 9% larger than that of the OmpC channel [48]. The OmpC porin showed a notably lower influx of ampicillin and benzylpenicillin than OmpF in *E. coli* because of the greater number of charged residues in the OmpC channel than in that of OmpF [50], and the lack of OmpF undoubtedly affects the efficiency of β-lactam hyposensitivity compared with the loss of OmpC [51]. As shown in previous studies, the hydrolysis rate of ΔOmpF OMVs was found to be lower than that of ΔOmpC OMVs against β-lactam antibiotics (Figure 6). As a result, when respective OMVs were added to susceptible *E. coli* in the presence of antibiotics, the group treated with ΔOmpC OMVs grew faster than the group treated with ΔOmpF OMVs (Figure 5). Moya-Torres et al. (2014) demonstrated that the deletion of *ompC* or *ompF* showed almost the same production of β-lactamase compared with the wild-type [51]. The lack of OmpF or OmpC did not induce intrinsic β-lactamase activity by the OMVs (Figure 4), indicating that the reduced hydrolysis efficiency of β-lactam antibiotics by OMVs was a result of the decreased permeability of β-lactam antibiotics due to loss of the porins (Figure 5 and Figure 6). Thus, our results indicate the crucial role of the porins in modulating the uptake of several β-lactam antibiotics into the lumen of OMVs, specifically, the influx of antibiotics is more efficient in the OmpF porin channel than the OmpC porin channel.

In summary, OMVs are important vehicles for substances related to β-lactam resistance, which help protect susceptible bacteria in the presence of β-lactam antibiotics. The mechanism of hydrolysis by OMVs against β-lactam antibiotics is not simply a one-protein effect, but rather an interaction between the β-lactamase in the lumen of OMVs and the porins on the surface of OMVs. The porin transports β-lactam antibiotics into the lumen of OMVs and β-lactamase in the lumen plays a key role in the direct degradation of the antibiotic. Our observation helps to elucidate the interaction of porins and β-lactamase in OMVs and increases our understanding of the resistance mechanisms found in multi-drug resistant bacteria.

## 4. Materials and Methods

### 4.1. Bacterial Strains

Antimicrobial-sensitive *E. coli* RC85 [7], antimicrobial-resistant *E. coli* RC85^+^ [7], and mutant RC85^+^ were used in this study. Bacteria were grown in Luria-Bertani (LB; Oxoid, Hampshire, UK) broth or LB agar. Broth cultures were grown at 37 °C with orbital shaking. Growth was monitored by measuring absorbance at 600 nm (OD_600_) using an xMark microplate spectrophotometer (Bio-Rad, München, Germany).

### 4.2. Molecular Cloning and Mutant Construction

Plasmid pRed/ET (amp) was obtained from the “Quick & Easy *E. coli* Gene deletion Kit” (Gene Bridges, Heidelberg, Germany) and the chloramphenicol resistance gene (Cm^R^) was amplified from pKINGeo/ccdB, which was designed in our laboratory [57]. An FRT-flanked, pro- and eukaryotic hygromycin selection cassette was obtained from “FRT-PGK-gb2-hygro-FRT template DNA” (Gene Bridges). The oligonucleotides (BIONEER, Daejeon, Korea) used in this study are listed in Table 2 and Appendix A. The pRed/ET (amp) vector was modified by inserting the Cm^R^ gene as a selection marker, because β-lactam-resistant *E. coli* RC85^+^ is resistant to ampicillin. Fragments 1 and 3 were amplified from pRed/ET (amp), while fragment 2 was amplified from pKINGeo/ccdB (Appendix A). Another round of PCR was performed to combine fragments 2 and 3 using respective primers, and the resulting amplicons were used as a template for the last round of amplification to attach fragment 1. The final DNA fragment flanked by Sac I and EcoRV was digested with Sac I/EcoRV and ligated into the Sac I/Msc I sites of pRed/ET (amp), forming the pRed/ET (Cm^R^). The “Quick & Easy *E. coli* Gene Deletion Kit” was used to construct the gene deletion mutant strains according to the manufacturer’s protocol, with some modifications [58]. The pRed/ET (Cm^R^) expression plasmid was transformed into the *E. coli* strain RC85^+^ by electroporation (Bio-Rad MicroPulser) at 1800 V with a 4 ms pulse rate. Transformants (RC85^+^ + pRedET) were selected on LB agar containing 35 μg/mL chloramphenicol (Sigma-Aldrich, USA) and grown overnight at 30 °C. A bacterial colony was selected from the plate and cultured in LB medium containing 35 μg/mL chloramphenicol overnight at 30 °C. Transformant cultures were re-incubated in super optimal broth (SOB) conditioned with L-arabinose (Sigma-Aldrich, St. Louis, MO, USA) at a final concentration of 0.3% (*w*/*v*) at 37 °C until an OD_600_ of 0.2 was obtained to induce pRedET. Induced cells were harvested by centrifugation for 30 sec at 16,000 × *g* in a cooled microfuge benchtop centrifuge and re-suspended in chilled 10% (*v*/*v*) glycerol. This process was repeated five times before electroporation. Competent RC85^+^ cells were mixed with generated hygromycin cassettes flanked by homology arms to replace the DNA fragment (Appendix A). Electroporation was performed with a Micropulser (Bio-Rad) delivering 1800 V for 4 ms. Electroporated transformants were immediately removed from the cuvettes by mixing with 1 mL LB medium without antibiotics and incubated at 37 °C for 3 h for recombination. Recombinant colonies were grown on LB agar containing 500 μg/mL hygromycin (Sigma-Aldrich) overnight at 37 °C for selection. Gene deletion mutants were confirmed through colony PCR using the sequencing primers (Table 2). PCR products were visualized on a 1% agarose gel and the band size was confirmed by comparing with the non-mutant *E. coli* (RC85^+^). Colonies from gene deletion mutants were identified by matrix-assisted laser desorption ionization-time of flight mass spectrometry (MALDI-TOF MS; Bruker Daltonik, Bremen, Germany) [59] to confirm that they were indeed *E. coli*.

### 4.3. Analysis of Antibiotic Resistance

Minimum inhibitory concentration (MIC) values were used to compare relative resistance levels of mutant strains to those of RC85^+^. Eight β-lactam antibiotics, namely amoxicillin, ampicillin, cefalexin, cefazolin, cefoperazone, cefotaxime, cloxacillin, and methicillin (Sigma-Aldrich) and five other classes of antibiotics, including amikacin, colistin, kanamycin, nalidixic acid, and streptomycin (Sigma-Aldrich) were selected for this. The MIC of each antimicrobial agent was determined using the broth-dilution method in 96-well plates [60] according to Clinical and Laboratory Standards Institute (CLSI) guidelines, except that cation-adjusted Muller Hinton broth was substituted with LB. The listed MIC values were presented as the mean of three independent experiments.

### 4.4. Isolation of Pure OMVs

Purification of OMVs was performed as previously described [7]. Briefly, the bacteria culture was centrifuged at 6000× *g* for 20 min, and the supernatant was filtered through 0.45-μm pore-sized vacuum filters. The filtered supernatant was concentrated by ultrafiltration using a QuixStand Benchtop system (GE Healthcare, Uppsala, Sweden). This was then centrifuged at 150,000× *g* at 4 °C for 3 h, and the OMVs purified on a continuous sucrose density gradient at 120,000 × *g* at 4 °C for 18 h. The OMV band was removed and centrifuged for 3 h at 150,000× *g* at 4 °C. The final OMV pellet was washed and resuspended in 10 mM Tris-HCl (pH 8.0) and filtered through a 0.2-μm filter. All purification steps were performed at 4 °C. The protein yields of OMV samples were measured using a Pierce BCA protein assay kit (Thermo Fisher Scientific, Foster City, CA, USA). Transmission electron microscopy (TEM) of OMVs was performed as previously described [7] using a Tecnai G2 Spirit Twin TEM system (FEI, Hillsboro, OR, USA). Dynamic light scattering (DLS) of OMVs for particle size distribution was performed as described previously [7] using a Nano ZS instrument (Malvern Instruments, Malvern, UK) and the Zetasizer software (version 7.11; Malvern Instruments).

### 4.5. Effect of OMVs on the Growth of Bacteria in the Presence of β-Lactam Antibiotics

The effect of OMVs on the growth of bacteria in the presence of β-lactam antibiotics was performed as previously described with slight modifications [7]. The effect of OMVs from RC85^+^, Δblc1 RC85^+^, ΔompC RC85^+^, and ΔompF RC85^+^ cells on the cytotoxicity of β-lactam antibiotics was monitored by assessing the growth of OMV-treated RC85 cells. The β-lactam antibiotics used were: penicillin family (ampicillin and amoxicillin), first-generation cephalosporin (cefazolin and cefalexin), and the third-generation cephalosporin (cefotaxime and cefoperazone). The following six antibiotics were used at concentrations known to inhibit RC85 growth: ampicillin, 30 μg/mL; cefotaxime, 1.25 μg/mL; cefoperazone, 4 μg/mL; amoxicillin, 12 μg/mL; cefazolin, 8 μg/mL; and cefalexin, 16 μg/mL. The MIC of β-lactam antibiotics against the RC85 were listed in Appendix A. Cultured RC85 cells (5 × 10^5^ CFU/mL) were inoculated into medium containing one of these antibiotics and 5 μg/mL of the respective OMV sample. RC85 in the antibiotic-free medium was used as a positive control, while the negative control consisted of bacteria and growth-inhibitory concentrations of the respective antibiotics. All tubes were incubated at 37 °C with shaking at 150 rpm. All experiments were performed in the dark to exclude the effect of light on the stability of the antibiotics used. The bacterial growth curves at OD_600_ were recorded at 3-h intervals up to 36 h using an xMark microplate spectrophotometer. Experiments were performed using three independent sets of bacterial cultures. The bacterial cultures were inoculated onto TSA with or without the respective same concentrations of antibiotics to confirm whether the susceptible bacteria could survive by antibiotic-resistant gene transfer via OMVs. Colonies from each cultured sample (*n* = 5, colonies per sample) on TSA without antibiotics were randomly selected and identified by MALDI Biotyper [59] to check contamination by other bacteria.

### 4.6. Quantification of β-Lactamase Activity

To test the differences in β-lactamase activity between OMVs from RC85^+^ and mutant strains, a colorimetric β-lactamase activity assay kit (BioVision, New Minas, NS, Canada) was used according to the manufacturer’s instructions. The assay is based on the hydrolysis of nitrocefin, a chromogenic cephalosporin producing a colored product that can be measured spectrophotometrically (OD_490_). A buffer of 10 mM Tris-HCl (pH 8.0) was used as a negative control and lyophilized positive control included in the kit was used. The quantity of enzyme capable of hydrolyzing 1.0 μM of nitrocefin/min at 25 °C corresponds to 1 U of β-lactamase. To liberate β-lactamase from the lumen of OMVs, each obtained OMVs were sonicated from 5 min (the effective sonication time on release β-lactamase from *E. coli*) [61], cooled on ice for 5 min [62], and centrifuged at 16,000× *g* at 4 °C for 20 min. Equal concentrations of each OMV sample (2.5 μg) were dispensed into the wells of a clear flat-bottomed 96-well, and nitrocefin and buffer (provided in the kit) were added to make a final volume of 100 μL. The absorbance at OD_490_ was immediately measured in kinetic mode for 60 min at 25 °C. For all measurements, three independent experiments were performed. A standard curve was generated using 0, 4, 8, 12, 16, and 20 nmol of nitrocefin, and the specific β-lactamase activity of each sample was expressed in milliunits/milligram of protein.

### 4.7. Measurement of Antibiotic Concentrations

Measurement of β-lactam antibiotic concentrations was carried out as previously described [7], with slight modifications. The effect of OMVs from RC85^+^ and mutants on the degradation of the six antibiotics listed above in a cell-free system were analyzed by liquid chromatography/electrospray ionization mass spectrometry (LC-ESI-QQQ-MS/MS; 6420 Triple Quad LC/MS; Agilent, Waldbronn, Germany). A 5 μg/mL sample of respective OMV in PBS was mixed with ampicillin (30 μg/mL), cefotaxime (1.25 μg/mL), cefoperazone (4 μg/mL), amoxicillin (12 μg/mL), cefazolin (8 μg/mL), or cefalexin (16 μg/mL). Filtered PBS containing the respective antibiotics without OMVs was used as a positive control. All samples were incubated at 37 °C with shaking at 150 rpm and diluted 20-fold prior to analysis. The concentrations of antibiotics were recorded at specific time points (ampicillin; 5 h, cefotaxime; 4 h, cefoperazone; 3 h, amoxicillin; 5 h, cefazolin; 1 h, and cefalexin; 11 h) in triplicate. For LC-MS/MS, LC-MS grade water (Burdick & Jackson, Muskegon, MI, USA) containing 5 mM ammonium formate (Sigma-Aldrich) and 0.1% formic acid (KANTO, Tokyo, Japan) (*v*/*v*) (solution A) and LC grade methanol (Burdick & Jackson) containing 5 mM ammonium formate with 0.1% formic acid (v/v) (solution B) were used as the mobile phase, at an initial A:B ratio of 30:70 or 50:50, depending on the antibiotic of interest. The compounds were separated using a Poroshell 120 EC-C18 column (2.1 × 100 mm, 2.7 μm; Agilent). Isocratic elution with phases A and B was followed by 3 min of total chromatography. The flow rate was 0.2 mL/min, the column temperature was 30 °C, and 99.99% pure nitrogen gas was used for desolvation. For the quantification of antibiotics, at least two or more transitions were selected for each analyte and the positive electric spray ionization (ESI+) was used with the multiple reaction monitoring (MRM) mode. The MassHunter software (version B.06.00; Agilent) was used to process the LC-MS/MS data and quantification of the analytes.

### 4.8. Statistical Analysis

Statistical analysis was performed using Graphpad Prism, version 8.1.1. (GraphPad, CA, USA). Significant differences were determined by One-way Analysis of Variance (ANOVA). Data are presented as mean ± standard deviation (SD). The difference was considered statistically significant at *p* < 0.05.

### 4.9. Data Availability

All data generated or analyzed during this study are included in this published article and its Appendix A.

## Figures and Tables

**Figure 1 ijms-21-02822-f001:**
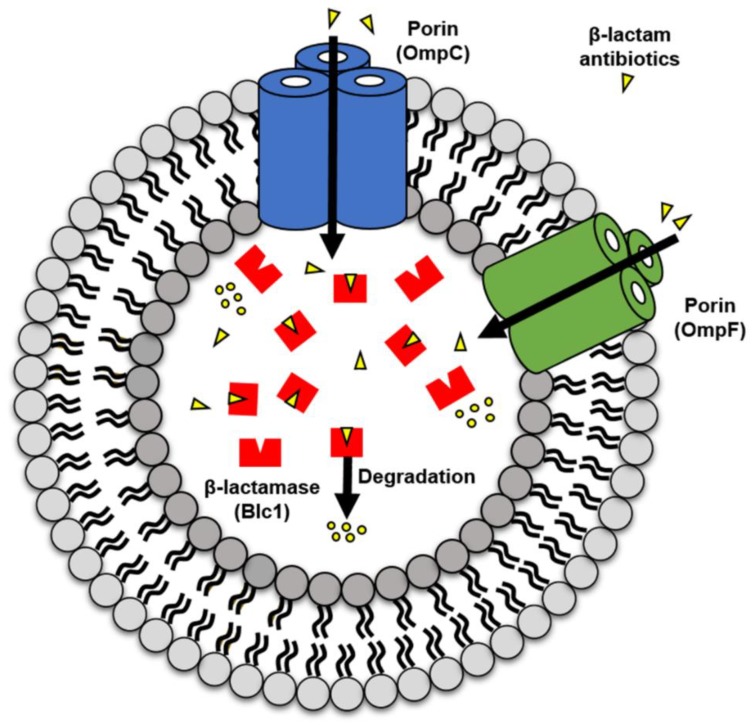
Predictive mechanism of β-lactam antibiotics degradation by outer membrane vesicles (OMVs) from β-lactam-resistant *Escherichia coli* (RC85^+^). OMVs take up β-lactam antibiotics into their lumen through porin channels (OmpC and OmpF) and the β-lactamase (Blc1) in the lumen hydrolyzes the β-lactam antibiotics confined in the lumen of OMVs.

**Figure 2 ijms-21-02822-f002:**
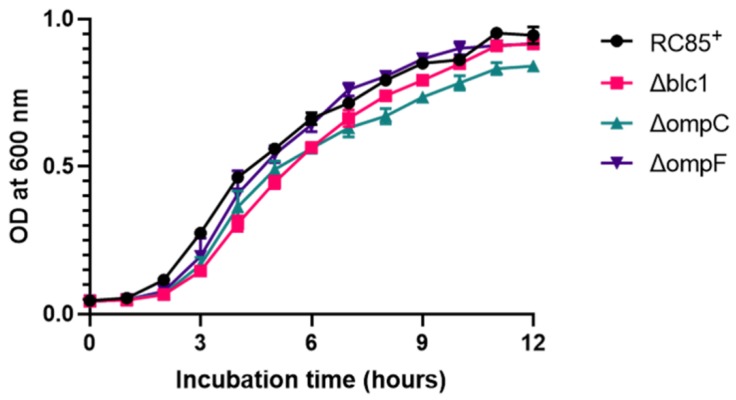
Growth curves of RC85^+^ and isogenic mutant strains of RC85^+^ (Δblc1, ΔompC, and ΔompF). The RC85^+^ and mutant strains were cultured on LB medium, and the growth of each strain was investigated by measuring absorbance at 600 nm. Data are presented as means and SEMs of three independent experiments.

**Figure 3 ijms-21-02822-f003:**
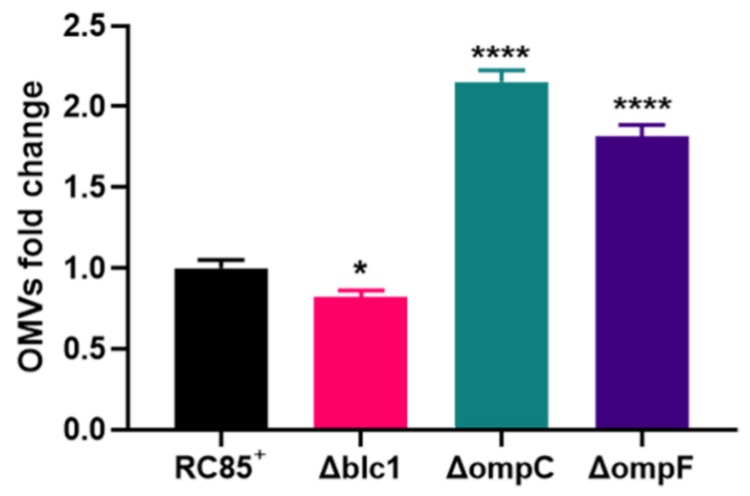
Production of OMVs isolated from RC85^+^ and isogenic mutant strains of RC85^+^ (Δblc1, ΔompC, and ΔompF). OMVs yields were averaged and normalized to RC85^+^ to adjust fold change. OMVs were purified and quantified using the BCA protein assay. Data are representative of three independent experiments in means ± SEMs. * *p* < 0.05, and **** *p* < 0.0001.

**Figure 4 ijms-21-02822-f004:**
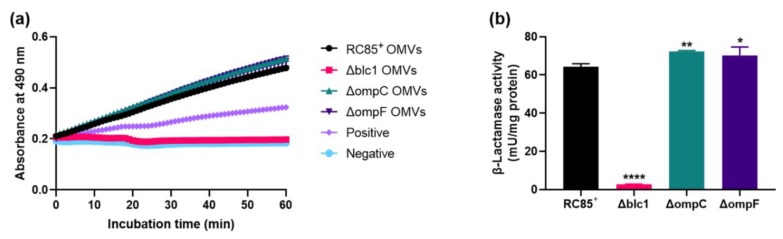
Investigation of the differences in β-lactamase activity between destroyed OMVs from RC85^+^ and isogenic mutant strains of RC85^+^ (Δblc1, ΔompC, and ΔompF). (**a**) β-Lactamase activity profiles of samples were measured in a kinetic mode in 60 min. Data are representative of three independent experiments in means ± SEMs. (**b**) β-Lactamase units were normalized to milligrams of total OMV protein. The data are presented as means and SEMs of three independent experiments. * *p* < 0.05, ** *p* < 0.01, and **** *p* < 0.0001.

**Figure 5 ijms-21-02822-f005:**
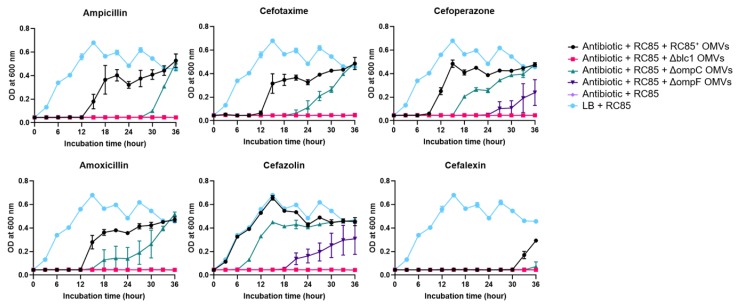
Growth of β-lactam-susceptible *Escherichia coli* (RC85) cells in antibiotic-induced growth inhibition environment to evaluate the antibiotic consumption role of intact OMVs from RC85^+^ and isogenic mutant strains. The growth-inhibiting concentrations of β-lactam antibiotics were: ampicillin, 30 μg/mL; cefotaxime, 1.25 μg/mL; cefoperazone, 4 μg/mL; amoxicillin, 12 μg/mL; cefazolin, 8 μg/mL; cefalexin, 16 μg/mL. The data are presented as means and SEMs of three independent experiments.

**Figure 6 ijms-21-02822-f006:**
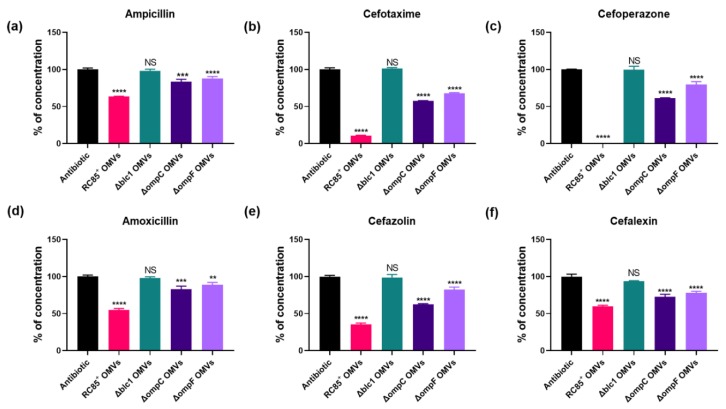
Evaluation of the concentration of β-lactam antibiotics incubated with 5 μg/mL of intact OMVs from RC85^+^ and isogenic mutant strains of RC85^+^ (Δblc1, ΔompC, and ΔompF). The initial concentrations and certain time points for measurement were as follows: ampicillin, 30 μg/mL, 5 h (**a**); cefotaxime, 1.25 μg/mL, 4 h (**b**); cefoperazone, 4 μg/mL, 3 h (**c**); amoxicillin, 12 μg/mL, 5 h (**d**); cefazolin, 8 μg/mL, 1 h (**e**); cefalexin, 16 μg/mL, 11 h (**f**). Respective antibiotics without OMVs were averaged and normalized as 100%, and the corresponding concentration of antibiotics with OMVs were calculated. The data are presented as means and SEMs of three independent experiments. The abbreviation ‘ns’ means not significant. ** *p* < 0.01, *** *p* < 0.001 and **** *p* < 0.0001.

**Table 1 ijms-21-02822-t001:** The MIC of β-lactam antibiotics and other class antibiotics against the multidrug-resistant *Escherichia coli* RC85^+^ and isogenic mutant strains of RC85^+^.

Class	Antibiotics	MIC (μg/mL)^a^			
		RC85^+^	Δblc1	ΔompC	ΔompF
β-lactam antibiotics	Ampicillin	>1024	4	>1024	>1024
	Cefotaxime	>1024	<1/2	>1024	>1024
	Cefoperazone	1024	<1/2	1024	>1024
	Methicillin	>1024	256	>1024	>1024
	Amoxicillin	>1024	2	>1024	>1024
	Cefazolin	1024	1	>1024	>1024
	Cefalexin	512	8	512	1024
	Cloxacillin	>1024	128	>1024	>1024
Other class antibiotics	Streptomycin	>1024	>1024	>1024	>1024
	Kanamycin	>1024	>1024	>1024	>1024
	Colistin	4	4	4	4
	Amikacin	8	8	8	8
	Nalidixic acid	>1024	>1024	>1024	>1024

^a^ MIC indicates minimum inhibitory concentration.

**Table 2 ijms-21-02822-t002:** Oligonucleotide sequence of primers and PCR product sizes.

Primer	Oligonucleotide Sequence (5′ to 3′)	Target Gene	Fragment Size (bp)
blc1-F	CTGGGTGTGGCATTGATTAAC	blc1	374
blc1-R	TAACGTCGGCTCGGTACG		
ompC-F	ATGAAAGTTAAAGTACTGTCCCTC	ompC	1103
ompC-R	TTAGAACTGGTAAACCAGACCC		
ompF-F	CTGACCGGTTATGGTCAGTG	ompF	599
ompF-R	CGTTTTGTTGGCGAAGCC		

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
