# Peer review of "The Importance of Porins and β-Lactamase in Outer Membrane Vesicles on the Hydrolysis of β-Lactam Antibiotics"

_ijms, 2020, doi:10.3390/ijms21082822_

Round 1

Reviewer 1 Report

Title: The importance of porins and beta-lactamase in outer membrane vesicles on the hydrolysis of beta-lactam antibiotics

This manuscript describes the ability of outer membrane vesicles (OMVs) from mutant E. coli to inactivate beta-lactam antibiotics and protect antibiotic-susceptible strains. This work addresses an important issue and is a logical extension of the authors' previous work. The results demonstrate that OMVs from mutants lacking beta-lactamase could neither degrade antibiotics nor protect susceptible bacteria. Outer membrane porins OmpC and OmpF appeared to be important in this process as well. Overall, the manuscript is clear and concise, and the experiments appear to have been performed conscientiously. However, there are several issues that need to be resolved.

Major revisions

  1. The most troubling aspect of this work is the inconsistency of results between different techniques. Namely, OMVs from ompC and ompF deficient strains exhibited superior beta lactamase activity (Fig. 4), yet were significantly less effective in protecting susceptible strains (Fig. 5) or in beta-lactamase activity as determined by mass spectrometry (Fig. 6). This discrepancy should be addressed, particularly as the authors claim "our results indicate the crucial role of the porins in modulating the uptake of several beta-lactam antibiotics into the lumen of OMVs". It may be that the composition/stability of the OMVs is also important, rather than just the presence or absence of ompC or ompF, which brings up point 2.
  2. No attempts were made to characterize the composition of OMVs. OMV numbers were inferred solely from protein concentration (Fig. 3). However, if a major outer membrane component such as ompC/F is absent, might this not alter the protein content of OMVs and skew the measurement of OMV numbers? Even a basic SDS-PAGE of OMVs from different strains might be nice to evaluate the relative amounts of Omp proteins from various mutant strains (see ref. 34).

Minor revisions

  1. Line 73: Consider starting a new sentence at ", and porins (OmpC and OmpF)..."
  2. Line 78: RC85+ should be defined
  3. Table 1: Some division between beta-lactam antibiotics and other classes should be evident. As it stands, streptomycin and kanamycin appear to be included with other class.
  4. Line 154: "grow" should be "growth"
  5. Line 167: "ability to hydrolysis" could be changed to "hydrolytic activity"
  6. Lines 250 and 251: To my knowledge, zwitterion refers to a compound with separate positive and negative charges, not two like charges. Consider changing to "dianionic".

Reviewer 2 Report

Comments for the Author:

Outer membrane vesicles (OMVs) produced by Gram-negative bacteria play significant and diverse biological roles including horizontal gene transfer, bacterial cellular stress responses, and resistance to phage and antibiotics. 

In this study, the authors investigate the role of OMVs from a drug resistant strain in conferring antibiotic resistance to an otherwise drug sensitive strain.  They examine OMVs isolated from a beta-lactam resistant strain and test whether certain factors in the OMVs influence the ability of these OMVs to protect against a range of beta-lactam antibiotics. The authors had previously shown that compared to OMVs from a susceptible strain, the OMVs from the resistant strain contained beta-lactamase enzyme and more porins. Here using a genetic approach, they show that the protective effect of these OMVs is dependent on the presence of the beta-lactamase (encoded by blc1). They also present data that suggests that porins (OmpF and OmpC) in OMVs play a role in the uptake of antibiotics into OMVs that facilitates their subsequent degradation by beta-lactamase.

Given that OMVs are produced by broad range of bacteria this study will be of interest to a large audience of microbiologists interested in OMVs and antibiotic resistance.

Specific comments:

  1. Why were MIC assays used to characterize the mutant strains in Section 2.1? Examination of growth in an antibiotic-induced growth inhibition environment similar to the OMVs (data shown in Figure 5) would enable a direct comparison to be made between the role of porins in whole cells versus OMVs. A growth assay is more sensitive than an MIC assay and might enable the authors to observe porin-dependent differences in beta-lactam sensitivity between the mutant strains which the authors state (Lines 103-106) was not possible with the MIC assay.

  1. In section 2.3 the authors present data to show that the beta-lactamase activity of the OMVs isolated from ompC and ompF mutants is similar to OMVs from the WT parent strain. The assay used to determine beta-lactamase activity, as described by the authors in Section 4.6,) involves incubation of OMVs with a nitrocefin substrate, a chromogenic cephalosporin that when hydrolyzed by beta-lactamase produces a colored product that can be measured. Presumably uptake of the nitrocefin into intact OMVs could be affected by the porins. It is not clear, as written, whether the OMV samples used were intact or whether the OMVs were disrupted to release their contents, including the beta-lactamase.

  1. In Fig. 6 a cell free-system was used to compare the ability of different types of OMVs to degrade antibiotics. What was the rationale for examining the antibiotics at various different timepoints (ranging from 1h to 11h). How were these time points chosen and why were they so varied?

  1. The authors had shown that OMVs from a RC85+ resistant strain contain an increased number of porins (Lines 182-184; Ref #7). Since OMVs are derived from the OM of bacteria it would follow that the OM of these resistant strains would also have increased porin levels. It seems counter-intuitive and an evolutionary disadvantage for a resistant strain to increase the number of porins which would enable more antibiotic to get in to the cell? Can the authors comment on how/why porin production is increased in these cells?

  1. Typo in Line 154: “did not show any grow” should be “growth”
  2. line 182: “porin protein” should be “porin proteins” or porins
  3. Lines 208, 215, 218: porin should be plural “porins”

Reviewer 3 Report

The authors have previously reported that the outer membrane vesicles (OMVs) derived from beta-lactam-resistant E. coli enable the survival of beta-lactam-susceptible E. Coli in the presence of beta-lactam antibiotics. In the current study, the authors present some interesting and novel findings regarding the mechanism of the OMV-mediated beta-lactam-resistance. However, there are concerns with the manuscript as a whole that must be addressed. Comment 1: The previous study showed the physical characterization of OMVs isolated from RC85+ and RC85 cells using TEM and DLS. The sizes and concentrations of the isolated OMVs from the mutant strains should be analyzed by TEM and DLS as well, not only by BCA.  Comment 2: Fig 4 is confusing. What positive and negative controls should be clearly defined. If positive control is the OMVs derived from RC85 strain, why is the absorbance at 490 nm of positive control increased? Comment 3: In Fig 5, the absorbance at 600 nm was monitored for 36 h. In the presence of the antibiotics, most of the strain showed their antibiotics resistance phenotype and started growing over ~12 h. It seems that most, if not all, of the OMVs would have been taken up by the E. coli in this time window, and such a slow effect does not fit the faster reaction that happened inside the isolated OMVs (Fig 4). The discrepancy of these reaction time frames should be discussed. Especially, OMVs were reported to transfer a variety of biomolecules, so the transfer of antibiotics resistance could be developed after the transfer of OMVs as well.  Comment 4: Related to the previous comment, the reduction of beta-lactam antibiotics incubated with OMVs from the mutant strains shown in Fig 6 was modest, compared to the striking effect of the transfer of antibiotics resistance phenotype shown in Fig 5. The data do not clearly show that the antibiotics resistance transfer is purely OMV-mediated or mediated by biomolecule transfer in the recipient E. coli.  Comment 5: Overall, the central hypothesis shown in Fig 1 seems to be oversimplified. 

Round 2

Reviewer 1 Report

The revised manuscript and the authors' response clarify several issues but also highlight several deficiencies in the current study. I now understand the data presented in figure 4, which is essentially a control to demonstrate that porin mutations do not influence overall beta-lactamase activity. Permeability of the actual drugs used is adequately addressed by Figure 6.

Regarding the second major issue, I appreciate the amount of effort that the authors invested into characterizing the OMVs but the results are inconclusive and of poor quality, as acknowledged by the authors themselves. I also realize that cost and personnel turnover are issues that all labs have to deal with. However, they are not relevant criteria for review. Although the basic premise of the paper is consistent with the results, the conclusions would be much stronger with a validation of OMV composition. That said, the revised paper is more clear and still addresses a topic of interest to readers.

Author Response

Comments and Suggestions for Authors

The revised manuscript and the authors' response clarify several issues but also highlight several deficiencies in the current study. I now understand the data presented in figure 4, which is essentially a control to demonstrate that porin mutations do not influence overall beta-lactamase activity. Permeability of the actual drugs used is adequately addressed by Figure 6.

Regarding the second major issue, I appreciate the amount of effort that the authors invested into characterizing the OMVs but the results are inconclusive and of poor quality, as acknowledged by the authors themselves. I also realize that cost and personnel turnover are issues that all labs have to deal with. However, they are not relevant criteria for review. Although the basic premise of the paper is consistent with the results, the conclusions would be much stronger with a validation of OMV composition. That said, the revised paper is more clear and still addresses a topic of interest to readers.

Answer:

We sincerely appreciate the reviewer’s critical comments. Based on your suggestions, we have improved our manuscript for the readers of this journal, IJMS. In the revised manuscript, we try to convey our research aims more effectively, emphasizing the key roles of porin proteins of OMVs in terms of the survival of bacteria in the presence of antibiotics. Our future study will focus on the specific roles of porin proteins in OMVs for the survival of bacteria, which may suggest overcoming antibiotic resistances of bacteria.

Reviewer 2 Report

The revised manuscript is much improved and authors adequately addressed the concerns raised in the initial review. 

Author Response

Comments and Suggestions for Authors

The revised manuscript is much improved and authors adequately addressed the concerns raised in the initial review.

Answer:

We would like to thank the reviewer for their initial comments, which have helped us to improve the quality of our manuscript.

Reviewer 3 Report

The objective of this paper with the title ‘The importance of porins and B-lactamase in outer membrane vesicles on the hydrolysis of B-lactam antibiotics’ was to understand the potential mechanism of the OMV-mediated defense system in bacteria. The authors addressed the concerns made in the previous review, but additional revision is required to improve the quality of the manuscript. 1. There are multiple grammatical errors in the revised manuscript (e.g. in line 75: we hypothesis, in line 114: exhibited the similar of OMVs, in line 126: since nitrocefin is uptake). 2. In fig. 4, the author clarified the OMVs were destroyed to release the encapsulated B-lactamases. To characterize the encapsulated enzyme activity it would be ideal to test intact OMVs. Can the authors also include the results of intact OMVs as well to strengthen the conclusion? 3. In line 156 - 163, the authors explained how they concluded that the OMV-mediated hydrolysis of B-lactam antibiotics was not due to OMV’s transfer of genetic materials. However, this paragraph needs more clarification for properly delivering the conclusion drawn from the results. 4. The authors provided TEM images of OMVs. However, in line 114-115, the resolution of the electron micrographs is not high enough to show the bi-layered spherical structure. Since the characterization of the isolated OMVs (e.g. size and morphology) is an essential part of this study, those data would strengthen the results. 

Round 3

Reviewer 3 Report

In the revised manuscript, most of the concerns of reviewers were appropriately addressed except for comment #2 regarding Fig. 4. The quality of the manuscript should not be compromised by the accessibility of the reagent if the authors agree that the experiment can reinforce the conclusion.    

Author Response

Comments and Suggestions for Authors

In the revised manuscript, most of the concerns of reviewers were appropriately addressed except for comment #2 regarding Fig. 4. The quality of the manuscript should not be compromised by the accessibility of the reagent if the authors agree that the experiment can reinforce the conclusion.    

Answer:

We apologize for not fully meeting the reviewer's comment in terms of Figure 4.

As shown in Figure 4, the activities of β-lactamase of OMVs from RC85+, ΔompC, and ΔompF were quite similar to each other. In contrast, the results shown in Figure 5 and Figure 6 indicated the significant differences in the capacity of β-lactam antibiotic consumption depending on the OMVs from RC85+, ΔompC, and ΔompF. These findings suggest that porins play an important role in antibiotic neutralization because the antibiotics are transported into the lumen of OMVs through the porin proteins. Since we demonstrated porin proteins are important for the neutralization of antibiotics in OMVs through Figure 5 and 6, it is sufficient to make the overall conclusion of this paper even if we do not put the control in Figure 4. We believe our conclusion will be reinforced if we follow the reviewer’s direction. Unfortunately, we do not have the reagents to perform the experiment currently, we could not add the experiments related to intact OMVs control. Please accept our apology for your disappointment.